# Gene-Mutation-Based Algorithm for Prediction of Treatment Response in Colorectal Cancer Patients

**DOI:** 10.3390/cancers14082045

**Published:** 2022-04-18

**Authors:** Heather Johnson, Zahra El-Schich, Amjad Ali, Xuhui Zhang, Athanasios Simoulis, Anette Gjörloff Wingren, Jenny L. Persson

**Affiliations:** 1Olympia Diagnostics, Sunnyvale, CA 94086, USA; heather@olympiadiagnostics.com; 2Department of Biomedical Sciences, Malmö University, SE-206 06 Malmö, Sweden; zahr.el-schich@mau.se (Z.E.-S.); anette.gjorloff-wingren@mau.se (A.G.W.); 3Department of Molecular Biology, Umeå University, SE-901 87 Umeå, Sweden; ali.amjad@umu.se; 4Department of Bio-Diagnosis, Institute of Basic Medical Sciences, Beijing 100005, China; zhanxuhui@inplorehealth.com; 5Department of Clinical Pathology and Cytology, Skåne University Hospital, SE-205 02 Malmö, Sweden; thanasis@simoulis.edu

**Keywords:** KRAS, colorectal cancer biomarkers, gene mutations, algorithm, colorectal cancer metastasis, colorectal cancer progression

## Abstract

**Simple Summary:**

Despite the high incidence and mortality of metastatic colorectal cancer (mCRC), there are no new biomarker tools available for predicting treatment response at diagnosis. We used machine learning using gene mutations from primary tumors of patients and developed a new biomarker model termed a 7-Gene Algorithm. We showed that this algorithm can be used as a biomarker classifier to predict treatment response with better precision than the current predictive factors. The 7-Gene Algorithm showed high accuracy to predict treatment response for patients suffering mCRC. The novel 7-Gene Algorithm can be further developed as a biomarker model for improvement of personalized therapies.

**Abstract:**

Purpose: Despite the high mortality of metastatic colorectal cancer (mCRC), no new biomarker tools are available for predicting treatment response. We developed gene-mutation-based algorithms as a biomarker classifier to predict treatment response with better precision than the current predictive factors. Methods: Random forest machine learning (ML) was applied to identify the candidate algorithms using the MSK Cohort (n = 471) as a training set and validated in the TCGA Cohort (n = 221). Logistic regression, progression-free survival (PFS), and univariate/multivariate Cox proportional hazard analyses were performed and the performance of the candidate algorithms was compared with the established risk parameters. Results: A novel 7-Gene Algorithm based on mutation profiles of seven KRAS-associated genes was identified. The algorithm was able to distinguish non-progressed (responder) vs. progressed (non-responder) patients with AUC of 0.97 and had predictive power for PFS with a hazard ratio (HR) of 16.9 (*p* < 0.001) in the MSK cohort. The predictive power of this algorithm for PFS was more pronounced in mCRC (HR = 16.9, *p* < 0.001, n = 388). Similarly, in the TCGA validation cohort, the algorithm had AUC of 0.98 and a significant predictive power for PFS (*p* < 0.001). Conclusion: The novel 7-Gene Algorithm can be further developed as a biomarker model for prediction of treatment response in mCRC patients to improve personalized therapies.

## 1. Introduction

Colorectal cancer (CRC) is one of the most prevalent cancers and a leading cause of cancer-related death globally [1,2]. Approximately 20% of patients at first diagnoses suffer metastatic CRC (mCRC), and another 25% will eventually develop metastatic disease in the US alone [3]. The 5-year survival among mCRC patients is below 20%, reflecting the poor prognosis of mCRC [3].

Currently, the established parameters, including tumor type, poor histological differentiation, and the depth of submucosal invasion, are used as prognostic factors for treatment response [4]. However, the high variability in the pathological assessment limits their clinical accuracy and contributes to the errors for decision making for tailored treatment and for predicting treatment outcome [5,6]. In recent years, the use of systemic treatments, such as first-line chemotherapy and adjuvant chemotherapy after surgical resection, has significantly improved the clinical outcome for distant mCRC patients [7]. However, biomarkers to predict chemotherapeutic efficacy and stratification of the patients who may benefit from adjuvant therapies are needed for personalized and optimal treatment for mCRC [4]. Clinical risk scores based on pathological and clinical parameters have been used for risk stratification of CRC patients [8,9,10]. Yet these predictive scores have limited application and have not been independently validated in clinical settings.

It is important to take into consideration of the high levels of heterogeneity and complexity of CRC, especially primary and metastatic lesions of mCRC harbor gain of function mutations in multiple oncogenes and loss of function in multiple tumor suppressors that are involved in proliferation, survival, and invasion [4,11]. Among the significantly mutated genes (n = 29) discovered in CRC patients, the most significantly mutated and well-known mutated genes in mCRC are the mutations in RAS and its related genes involved in cancer cell proliferation, survival, and invasion pathways [12]. The mutations in KRAS and NRAS frequently occur in primary CRC tumors, with 36% for KRAS and 3% for NRAS [13]. KRAS is a major component of the mitogen-activated protein kinase pathway, which can be activated by a ligand binding to epidermal growth factor receptor (EGFR). EGFR treatment resistance is mediated by mutations in KRAS, which lead to constitutive activation of the RAS–RAF–MEK–ERK pathway. In total, 85–90% of patients with mutations in KRAS codons 12 and 13 (exon 2) may develop resistance to EGFR therapies, such as cetuximab and panitumumab [14]. Due to the fact that mCRC with wild-type KRAS or NRAS do not respond to anti-EGFR therapies, this leads to a hypothesis that additional mediators may be involved, such as BRAF, ERBB2 (HER2), and microsatellite instability (MSI) [15]. BRAF mutation is found in 8–12% of mCRC and confers poor prognosis. BRAF mutation can activate the MEK/ERK pathway, which increases cell proliferation and inhibits apoptosis [16]. In addition, it is shown that BRAF mutations may predict EGFR treatment resistance and such mutations do not overlap with RAS mutations [17]. ERBB2 gene encodes human epidermal growth factor receptor 2 (HER2), which is a key gene associated with poor prognosis and drug resistance in mCRC [12,18]. Alterations in genes associated with PI3K/Akt/PTEN/mTOR pathway are also frequently observed in CRC, and PI3K/Akt/PTEN/mTOR pathway is under the control of the RAS activity [15]. TSC2 is a downstream target of Akt phosphorylation, and it forms a complex with TSC1 to suppress mTOR activity. Mutations of TSC2 occur in some CRC patients [14]. TP53 mutations occur in over 50% of CRC, and different TP53 mutations may have different effects on patient survival [19]. Mutations of APC are also associated with poor overall survival in mCRC patients irrespective of RAS and BRAF status [20]. Although multiple mutations in genes encoding the proteins from the cascades of RAS–RAF–MEK ERK, PI3K/Akt/PTEN/mTOR, P53, and APC pathways are important driver mutations to promote cancer progression and treatment resistance, there is a lack of reliable biomarker models based on the combination of the multiple gene mutations to predict CRC metastasis and treatment response [4].

Machine learning (ML)-based algorithms and models developed by using CT or MR imaging and tissue morphology on sections are becoming useful in clinical decision making. Of these, the least absolute shrinkage and selection operator (LASSO) is one of the algorithms and its clinical efficacy has been demonstrated previously in predicting LNM in T1 CRC [21,22]. However, given that the surgical resected tumor biopsies after endoscopic resection exhibit morphological heterogeneity and complexity in both cancer cells and cancer-associated microenvironment, these factors limit the accuracy of the ML algorithms that are mainly dependent on the quality of the imaging and tissue sections.

Currently, ML-based prediction models have emerged as powerful tools for predicting disease metastasis and treatment response in CRC. ML techniques have been applied to analyze individual target lesions of patients with mCRC, and to investigate mCRC patient outcome after cetuximab treatment [23]. ML-based cancer prediction models using molecular and genomic profiles are beneficial for patients with stage II–III CRC [24]. These studies suggest that ML models have comparable predictive power for determining cancer recurrence in subgroups of CRC patients. Recent studies reported that the frequency of mutations in KRAS, APC, KIT, FBXW7, SMAD4, PTEN, and CDKN2A genes was numerically higher in primary tumors than in metastatic CRC lesions [25]. The new approaches by using an ML-based decision support system (DSS), combined with random optimization (RO), have been applied to extract clinical information from breast cancer patients. These new approaches have demonstrated that implementation of ML algorithms and RO models into clinical data classification may have the potential to revolutionize the practice of personalized medicine [26]. More recently, an ML-based decision tree model was used for predicting adoption of CRC screening among Korean Americans, suggesting that ML techniques have great impact on social- and health-care systems [27]. 

There is a rapid development in new technologies enabling to obtain large amount of genomic, epigenomic and imaging data from primary tumors of each individual patients, artificial intelligence ML-based tools are especially useful not only for data processing but also for early detection and prognostics of cancer. 

We have previously developed and used an ML-algorithms-based biomarker panel using gene expression profiles from primary tumor specimens and urine of large cohorts of prostate cancer patients [28,29,30]. However, there is no systematic implementation of ML algorithms-based on gene-mutation profiles for prediction of treatment response of mCRC. In this study, we therefore aimed to apply our established ML methods to develop a gene-mutations-based algorithm for prediction of treatment response in CRC. We used two cohorts as training and validation sets. We assessed the performance of our newly identified algorithm as a biomarker classifier to predict treatment response of patients with CRC and mCRC. Our findings suggest that the 7-gene algorithm may be developed as a predictive biomarker for improvement of personalized therapies and to reduce mortality in clinical practice. 

## 2. Materials and Methods

### 2.1. CRC Patients Cohorts

For the colorectal cancer MSK cohort, the data of 471 patients with unresectable colorectal cancer (CRC) treated at Memorial Sloan Kettering were obtained from cBioportal [31,32]. Gene alterations, mutations, genomic profiling, and clinical data including diagnostic age, cancer stage, microsatellite instability (MSI), metastasis, prior adjuvant therapy status, prior surgery on primary tumor, progression after first-line chemotherapy, and overall survival during follow-up were extracted [31]. Out of 471 CRC patients, 388 had distant metastasis during a 50-month follow-up period. The patient demographics and clinical characteristics are detailed in Table 1.

For the colorectal cancer cohort of The Cancer Genome Atlas (TCGA) Firehose Legacy, all genomic and clinical data were extracted from cBioportal (https://www.cbioportal.org/ (accessed on 30 December 2021)). Two databases on colorectal cancer were searched and obtained from cBioportal (30 December 2021). A total of 191 out of 221 patients with gene mutation and information on cancer progression/recurrence after treatment during follow-up formed the TCGA Cohort. Among these patients, 32 developed distant metastases during a 50-month follow-up period. The patient demographics and clinical characteristics are detailed side-by-side with that of the MSK cohort in Table 1. This retrospective analysis was approved by the Swedish Ethics Authority.

### 2.2. Algorithms for Prediction of Cancer Progression after Treatment

A random forest machine learning algorithm screening was performed to select combinations of mutation profiles of the genes in the RAS–RAF–MEK–ERK and PI3K/Akt/PTEN/mTOR pathways, as well as TP53 and APC, which are frequently mutated in CRC, to form classifiers by using the established methods previously described in [28,29,30]. Using the MSK cohort as a training set, the random forest algorithm classifiers, which combine different gene mutation profiles, were used to distinguish progression and non-progression using XLSTAT (Addinsoft, Paris, France). For development of each random forest algorithm, the size of the forest was determined by the number of patients in the cohort (>½ of the patient number). Each tree was developed from a bootstrap sample selected from the training data, with an arbitrary subset of genes being drawn. Confusion matrix of each random forest algorithm to show accuracy for classification of progression and non-progression was used to identify the algorithm with the highest classification accuracy. Further, 10-fold cross-validation was conducted on high-performing gene mutation combinations in a grid search to verify the classification performance and find the best gene combination. Among the algorithms of the gene combinations tested, a 7-Gene Algorithm consisting of KRAS, BRAF, ERBB2, MAP2K1, TSC2, TP53, and APC showed the highest accuracy to distinguish progressed and non-progressed patients after treatment and was chosen as the classifier. With this 7-gene set, the random forest parameters, such as the number of trees and node size, were further tuned to optimize the accuracy and formed the final algorithm for classification of progression and non-progression.

### 2.3. Statistical Analysis

To assess the predictive accuracy for progression after treatment, logistic regression analysis was performed to compare progression predicted by the 7-Gene Algorithm with progression status after treatment during follow-up for each sample to calculate sensitivity, specificity, positive predictive value, negative predictive value, and their respective 95% confidence intervals (CI) in XLSTAT (Addinsoft, Paris, France). To ensure a fair comparison of the models, we used the receiver operating characteristic (ROC) curve, area under the curve (AUC), sensitivity (recall), specificity, accuracy, average precision (AP), false positive rate, and precision as performance indicators. We used the AU-ROC as the performance index and the AP value as the criterion for the precision–recall (PR) curve. In addition, discriminant analysis was conducted to test the predictive accuracy and the result was compared with that from logistic regression as described previously [28,29]. Similarly, the predictive accuracy for cancer stage, and status of prior adjuvant therapies, surgery on primary tumor and microsatellite instability (MSI), and their combination with the 7-Gene Algorithm was assessed using logistic regression analysis.

To determine the predictive power for cancer progression after treatment, univariate and multivariate Cox proportional hazards regression analyses and Kaplan–Meier survival plot of progression-free survival for the 7-Gene Algorithm, cancer stage, and status of neoadjuvant therapies, surgery on primary tumor and MSI were conducted using XLSTAT. Dot plots were created to show the distribution of the classification score of individual samples in the non-progressed and progressed patients after treatment in the MSK cohort and TCGA cohort using Graphpad (GraphPad Software, San Diego, CA, USA). The nonparametric Mann–Whitney test was performed to compare the patient groups using XLSTAT.

## 3. Results

### 3.1. Development of the 7-Gene Algorithm for Stratification of Responder and Nonresponder Patients to Predict Response to Treatment

Since mutations of genes in the RAS–RAF–MEK–ERK and PI3K/Akt/PTEN/mTOR pathways, as well as TP53 and APC, are predominantly involved in CRC treatment response [4], we wanted to examine whether the mutation profiles of the genes in these pathways may be used to predict treatment response. Disease progression after treatment is a major indicator of treatment response; we, therefore, examined if a gene-mutation-based ML model might be developed as biomarkers to stratify and predict treatment response of CRC patients at diagnostic occasions (Figure 1). Based on the clinical data of 447 patients in the MSK cohort, we divided patients into two subgroups: (i) the responder group: patients had no disease progression after first-line chemotherapy during 50 months; (ii) the non-responder group: patients experienced disease progression after first-line chemotherapy during a 50-month period. We then utilized a random forest machine learning classification screening to test if various combinations of mutation profiles of the candidate genes might be able to distinguish responders from nonresponders. An algorithm termed 7-Gene Algorithm consisting of mutation profiles of the seven genes: KRAS, BRAF, ERBB2, MAP2K1, TSC2, TP53, and APC exhibited the highest accuracy for classification compared with all other gene-mutations-based algorithms tested, as determined using logistic regression analysis. The 7-Gene Algorithm had sensitivity of 83% (95%CI: 68–98%), specificity of 98% (95% CI: 97–100%), and the accuracy of performance AUC of 0.98 (95% CI 0.95–1.02) to distinguish responders from non-responders (*p* < 0.001; Table 2, Figure 2A). We compared the accuracy of performance between the 7-Gene Algorithm and the clinical and pathological risk indicators, including cancer stage, adjuvant therapies, surgery on primary tumor, and MSI. Logistic regression analysis revealed that the utility of cancer stage to distinguish responders from non-responders had AUC value of 0.5 (Table 2, Figure 2B). The adjuvant therapies had 0% sensitivity and AUC of 0.41. Similarly, surgery on primary tumor had 0% sensitivity and AUC of 0.41. MSI had 0% sensitivity and AUC of 0.34 (Table 2, Figure 2C–E). When the 7-Gene Algorithm was combined with all of these parameters together, cancer stage, adjuvant therapies, surgery on primary tumor and MSI, the sensitivity and AUC values remained similar to that of the 7-Gene Algorithm alone (Table 2, Figure 2F). These data showed that the 7-Gene Progression Algorithm had statistically significant accuracy as a classifier to distinguish responder and non-responder patients to the first-line chemotherapy; however, there was no statistical significance when using the clinical and pathological indicators, including cancer stage, adjuvant therapies, surgery on primary tumor, and MSI, as classifiers to stratify the subgroups of patients.

### 3.2. Assessment of the 7-Gene Algorithm for Prediction of Progression-Free Survival after Treatment in the MSK Cohort

To assess whether the 7-Gene Algorithm might be used as a biomarker to predict progression-free survival (PFS) in the MSK cohort, the log-rank analysis was performed. Kaplan–Meier plot with log-rank analysis revealed that there was a statistically significant difference in PFS between the subgroups stratified based on 7-Gene Algorithm scores. Patients with high scores of the 7-Gene Algorithm in their primary tumor at diagnosis had significantly poorer PFS compared with those with low scores (*p* < 0.001, Figure 3A). Next, we examined whether the clinical and pathological indicators, including cancer stage (stage I/II vs. III/IV) and adjuvant therapies (therapy vs. no therapy), surgery on primary tumor (surgery vs. no surgery), and MSI type (stable vs. instable), may be used to predict PFS in the MSK cohort. Kaplan–Meier plot with log-rank analysis revealed that there was no statistically significant differences in PFS between subgroups stratified based on the status of cancer stages, therapies, and MSI type (for cancer stage, *p* = 0.125; for neoadjuvant therapies, *p* = 0.876; and for MSI type, *p* = 0.093; Figure 3B,C,E), while there was a small but statistically significant difference between the subgroups stratified based on the surgery status on primary tumors (*p* = 0.012, Figure 3D).

As comparison to the algorithm, we examined whether the mutation status of each individual gene in the 7-Gene Algorithm: KRAS, BRAF, MAP2K1, ERBB2, TSC2, APC, and TP53 may be used to predict PFS. We performed Kaplan–Meier analysis to compare PFS of patients who have mutant vs. those who have wild type of each gene in their primary tumors determined at the diagnosis. There was no statistically significant difference in PFS between the mutant and WT groups stratified based on each of the individual gene mutation status: KRAS, MAP2K1, ERBB2, TSC2, and TP53 genes (for KRAS, *p* = 0.857; for MAP2K1, *p* = 0.584; for ERBB2, *p* = 0.951; for TSC2, *p* = 0.982; for TP53, *p* = 0.772). Meanwhile, there was a statistically significant difference between the patients with mutant of BRAF or APC and those with WT of these individual genes in their primary tumors (for both BRAF and APC, *p* < 0.001) (Appendix A). These data suggest that the 7-Gene Algorithm might be used as a biomarker to predict progression-free survival (PFS) with better precision as compared with each individual gene in the MSK cohort.

A dot plot analysis was further performed to illustrate the distribution of the classification scores of the 7-Gene Algorithm between the treatment responder and non-responder patients in the MSK cohort. The plot showed a statistically significant difference in the 7-Gene Algorithm scores between the two patient groups (*p* < 0.001, Figure 4). Taken together, the results from logistic regression analyses, Kaplan–Meier plot, and dot plot were consistent and suggesting the accurate performance of the 7-Gene Algorithm as a biomarker for predicting treatment response.

### 3.3. The 7-Gene Progression Algorithm for Prediction of Progression after Treatment

To further assess whether the 7-Gene Algorithm may be used as an independent predictive biomarker to predict the treatment response of CRC at first diagnostic occasion, we performed univariate and multivariate Cox proportional hazard regression analyses based on PFS in the MSK cohort. The univariate analysis revealed that the prediction power of the 7-Gene Algorithm for PFS, as indicated using the hazard ratio (HR), was 7.5 (95% CI: 3.5–15.9, *p* < 0.001; Table 3). While the HR value for cancer stage was 1.3 (95%CI: 0.9–1.9, *p* = 0.128), HR for adjuvant therapy was 1.1 (95% CI: 0.8–1.3, *p* = 0.877), HR for surgery was 0.8 (95% CI: 0–1.0, *p* = 0.013), and HR for MSI was 0.7 (95% CI: 0.5–1.1, *p* = 0.097; Table 3). These data show that the 7-Gene Progression Algorithm has much higher HR and is statistically significant in predicting PFS as compared to the other clinical and pathological indicators. To further confirm the predictive value of the 7-Gene Algorithm for PFS in relation to the clinical indicators, we performed multivariate Cox analysis. Interestingly, the HR of the 7-Gene Algorithm to predict PFS as an independent biomarker was 8.9 (95% CI: 4.0–20.1, *p* < 0.001), whereas the HR for cancer stage was 1.1 (95% CI: 0.7–1.5, *p* = 0.75), HR for adjuvant therapy was 1.1 (95% CI: 0.8–1.4, *p* = 0.536), HR for surgery was 0.7 (95% CI: 0–0.9, *p* = 0.002), and HR for MSI was 0.6 (95% CI: 0–0.9, *p* = 0.009; Table 3). These results suggest that the 7-Gene Algorithm has great potential to be used as a predictive biomarker for PFS.

### 3.4. Validation of the 7-Gene Algorithm in the TCGA Cohort

To validate the 7-Gene Algorithm for prediction of progression after treatment, a TCGA cohort with 119 patients was used (Figure 1 and Table 1). In this cohort, 30 out of 119 patients responded to treatment with no progression/recurrence. The same random forest machine learning algorithm using the mutation profiles of the seven genes as developed in the MSK cohort was used to classify each patient as treatment responder without progression or treatment non-responder with progression. Logistic regression analysis revealed that the 7-Gene Algorithm exhibited high accuracy in distinguishing responder and non-responder patient groups with sensitivity of 96% (95% CI 93–99%), specificity of 77% (95% CI 62–92%), and AUC of 0.97 (95% CI 0.95–0.99) (*p* < 0.0001) (Table 2, Figure 5A). These data were similar to what was obtained by using the MSK cohort. Similar to what was observed in the MSK cohort, the clinical and pathological parameters, including cancer stage, neoadjuvant therapies, surgery, and MSI, did not exhibit high values of specificity and high AUC values in distinguishing responders and non-responders (Table 2, Figure 5B,C). The logistic regression analysis was performed by using the 7-Gene Algorithm in combination with all the above-mentioned clinical indicators. The data showed that the performance of the 7-Gene Algorithm together with all the clinical indicators remained similar to that of the 7-Gene Algorithm alone to distinguish responders and non-responders to treatment in the TCGA cohort (Table 2, Figure 5D).

To further validate the performance of the 7-Gene Algorithm as a predictive biomarker for treatment response, Kaplan–Meier analysis was performed using the TCGA cohort. Similar to what was observed using the MSK cohort, patients with high scores of the 7-Gene Algorithm in their primary tumor at diagnosis had significantly poorer PFS compared with those with low scores (*p* < 0.001, Figure 6A). In this cohort, only two clinical and pathological indicators, cancer stage and adjuvant therapies, were available. We examined whether the clinical and pathological indicators, including cancer stage (stage I/II vs. III/IV) and adjuvant therapies (therapy vs. no therapy), may be used to predict PFS in the TCGA cohort. Kaplan–Meier plot with log-rank analysis revealed that there were no statistically significant differences in PFS between subgroups stratified based on the status of cancer stages or therapies (for cancer stage, *p* = 0.75; for adjuvant therapies, *p* = 0.72; Figure 6B,C).

As a comparison, the ability of the mutation status of each of the seven genes in the algorithm to predict PFS was also assessed by Kaplan–Meier plot in the MSK and TCGA cohort. Mutations in KRAS, ERBB2, TSC2, and TP53 had no statistical significance in PFS in this cohort (*p* = 0.122 for KRAS, *p* = 0.774 for BRAF, *p* = 0.162 for ERBB2, *p* = 0.474 for TSC2, *p* = 0.081 for APC, *p* = 0.948 for TP53, Appendix A). There was a significant difference in PFS between patients with WT MAP2K1 and MAP2K1 mutation (*p* < 0.001, Appendix A). The result showed that the majority of the individual gene mutations did not exhibit the statistical significance to stratify patients’ PFS.

To further validate the performance of the 7-Gene Algorithm as a predictive biomarker for treatment response in the TCGA cohort, we performed univariate and multivariate Cox regression analyses in the TCGA validation cohort. In the univariate analysis, the predictive power for PFS as indicated by HR of the 7-Gene Algorithm was 16.9 (95% CI 7.2–39.6) (*p* < 0.001), whereas the HR for cancer stage was 1.2 (95% CI 0.5–2.7, *p* = 0.723) and HR for adjuvant therapies was 3.0 × 10^−7^ (95% CI 0-Inf, *p* = 0.997) (Table 3). In the multivariate analysis, HR of the 7-Gene Algorithm was 16.9 (95% CI 7.2–39.7) after adjusting for cancer stage and adjuvant therapies (*p* < 0.001) (Table 3), which was similar to that in the univariate analysis. HR value for cancer stage and adjuvant therapies was also similar in the univariate analysis (Table 3). Interestingly, the HR values of the 7-Gene Algorithm to predict PFS were higher in the TCGA cohort than those in the MSK cohort. Similar to what was observed in the MSK cohort, the dot plot in the TCGA cohort showed statistically significant difference in the 7-Gene Algorithm classification scores between the treatment responder and non-responder patients (*p* < 0.001, Figure 7). This further showed the ability of the 7-Gene Algorithm to distinguish progressed and non-progressed patients. All of the assessment results in the TCGA cohort were consistent with those obtained in the MSK cohort and confirmed the high accuracy of the 7-Gene Algorithm for prediction of cancer progression after treatment.

### 3.5. Assessment of the 7-Gene Algorithm for Prediction of Treatment Response in mCRC Patients

Out of 471 CRC patients, 388 patients had metastatic disease in the MSK cohort (Table 1). In clinical practice, there is no predictive biomarker available to predict treatment response for mCRC patients. We, therefore, wanted to examine whether the 7-Gene Algorithm may be used to predict response for these 388 patients with mCRC. Kaplan–Meier plot with log-rank analysis was performed and revealed that there was a statistically significant difference in PFS between the subgroups stratified based on 7-Gene Algorithm scores in the mCRC cohort. mCRC patients with high scores of the 7-Gene Algorithm in their primary tumor at diagnosis had significantly poorer PFS compared with those with low scores (log-rank *p* < 0.001; Figure 8A). Similar to what was observed in the total population of the MSK cohort, there was no statistically significant difference in PFS between the subgroups stratified by using pathological indicators, including cancer stage (stage I/II vs. III/IV) and adjuvant therapies (therapy vs. no therapy) (for cancer stage, *p* = 0.190; for adjuvant therapies, *p* = 0.669) (Figure 8B,C). Meanwhile, there was a small but statistically significant difference in PFS between the subgroups stratified by using surgery on primary tumor (surgery vs. no surgery) and MSI type (stable vs. instable) (for surgery, *p* = 0.043, for MSI type, *p* < 0.001, Figure 8D,E).

To further assess whether the 7-Gene Algorithm may be used as an independent predictive biomarker to predict treatment response of mCRC patients at the diagnostic occasion, we assessed the predictive value of the 7-Gene Algorithm as an independent biomarker for PFS of mCRC patients by using univariate and multivariate Cox proportional hazard regression analyses. The univariate analysis revealed that the prediction power of the 7-Gene Algorithm for mCRC PFS as indicated using HR was 16.9 (95% CI 4.2–68.0, *p* < 0.001). The multivariate analysis revealed the prediction power of the 7-Gene Algorithm for mCRC PFS with an HR of 17.6 (95% CI 4.4–70.8, *p* < 0.001) in relation to cancer stage (stage I/II vs. III/IV), adjuvant therapies (therapy vs. no therapy), surgery on primary tumor (surgery vs. no surgery), and MSI type (stable vs. instable), and none of these clinical indicators exhibit statistical significance as predictive biomarkers for PFS in mCRC patients using univariate and multivariate Cox analyses (Table 4). Interestingly, the predictive HR values of the 7-Gene Algorithm for predicting PFS in mCRC patients were much higher than its predictive HR values in the total population of the MSK cohort as determined using univariate and multivariate analyses. Our data suggest that the 7-Gene Algorithm may be used as a predictive biomarker for stratifying and predicting treatment response of mCRC patients at the first diagnostic occasions.

## 4. Discussion

Some new diagnostic and prognostic biomarkers have been developed based on the molecular pathological parameters, including the microsatellite instability-high (MSI-H), mismatch repair-deficient (MMR-D), and mutations in KRAS, NRAS, and BRAF genes [15]. These biomarkers are useful in clinical decision making for targeted treatment of mCRC. However, these individual biomarkers did not exhibit high sensitivity and specificity to be used to stratify subgroups with risk of metastatic disease and to accurately predict treatment response.

In this study, we described the performance of our newly developed ML-based algorithms in a side-by-side comparison with the clinically well-known biomarkers in stratification and prediction of response to treatment in two CRC cohorts. We demonstrated that the 7-Gene Algorithm developed in this study exhibited significantly accurate performance as biomarkers to distinguish subgroups of CRC patents with a high risk of not responding to treatment. The novelty of our finding in this study is that we developed the 7-Gene Algorithms based on the screening of the well-known mutations and build a model to combine the multiple gene mutation profiles as a single biomarker by using the ML method to predict treatment response of patients with CRC, in particular, patients with mCRC.

There is an urgent need to develop multi-module prediction biomarkers/models by using advanced technologies to improve precision medicine and reduce CRC-caused mortality. The high variability in the pathological assessment has limited the performance of the parameters to identify and distinguish patients into subgroups and predict their individually based response to treatment; we took the advantages of the ML and developed the prediction biomarker models based on the multiple gene mutations, which showed high accurate performance. Since the 7-Gene Algorithm prediction model is derived from the complexity of the cancer genome, in which multiple mutations in cancer-specific pathways co-exist, this algorithm exhibited the higher accuracy as determined using AUC as compared to clinical parameters, including cancer stage at diagnosis, adjuvant therapies and surgery on primary tumor, and MSI status. Currently, no biomarkers are available to stratify CRC patients suffering metastatic disease or that have risk of developing cancer metastasis at the diagnostic occasions. It has been shown that, among liver mCRC patients, 20–25% was resectable despite the use of novel diagnostic and therapeutic methods, 60–70% of distant mCRC patients develop local or distant recurrence, while only 20% achieve long-term remission [33,34,35].

In this study, we showed that the 7-Gene Algorithm has statistically significant power to predict DFS in the mCRC cohort at the diagnostic occasion. As it is known that the gene mutations of KRAS, BRAF, TP53, and PI3K play key roles in progression of cancer cell invasion and metastasis, the algorithm based on the combinational mutation profiles of these key genes may reflect the nature of cancer metastasis. The single gene mutation of KRAS, BRAF, TP53, and PI3K has been tested for stratifying resectable and unresectable mCRC patients with varying results [36]. Two recent large mutational studies on the large cases of mCRCs have been conducted. Genomic profiling provides an overview of the genomic landscape of mCRC in a single analysis, including actionable targets and markers of immune sensitivity. Amplification of ERBB2 was present in 1% of cases. MSI-H status was reported in 3%, and 38% of them also harbored the BRAF V600E mutation. These studies suggest that there is a clear advantage of comprehensive genomic profiling techniques over a single gene’s mutation data in tumors, and that the comprehensive genomic profiling techniques can provide extensive information about the tumor molecular landscape [37]. In recent CRC clinical FIRE-3 trials, RAS, BRAF V600E, and SMAD4 mutations were identified as poor prognostic biomarkers in patients, whereas improved outcome for cetuximab efficacy was observed for BRAF non-V600E mutation [38].

Our results showed that each of the 7 genes had no or low statistically significant power to predict PFS in the two CRC cohorts used in this study. Our finding suggests that the ML-based algorithms provide future direction and modern tools for improvement of diagnostics and prognostics. However, none of the gene mutation tests have achieved high prognostic accuracy and reliability to justify clinical application [4].

ML-based algorithms have emerged as useful and modern tools in diagnostics and prognostics in CRC. However, the current ML models are developed based on the histopathological data, such as lymphovascular invasion, tumor budding, and precise depth of tumor invasion. The 7-Gene Algorithms in this study has advantages due to the fact that the cancer genomic data are not affected by the pathological observations as shown in this study. Moreover, the molecular mechanisms and functional studies in cell-line-based and animal-based models have shown that these genes control the cancer metastasis and treatment resistance in mCRC. We have further shown that the 7-Gene Algorithms showed consistently high predictive accuracy in two independent cohorts, suggesting that the robust and reliable features of using the gene mutation profiles in algorithms.

Although the 7-Gene Algorithms showed high performance as predictive biomarkers, there are some limitations in this study. First, we do not have prospective cohorts and also lack large cohorts to validate the two algorithms. In the future, more studies will be conducted in large patient populations to further validate the two 7-Gene Algorithms for prediction of mCRC and cancer progression. To further improve the accuracy of the established model, it is necessary to further conduct studies in large patient cohorts to improve CRC treatment and reduce mortality in clinical practice.

## 5. Conclusions

In conclusion, we established and compared the 7-Gene Algorithm side-by-side with the available clinical and histopathological indicators to predict treatment response in CRC. This biomarker model has great advantages to be further developed and validated in large patient cohorts. Utility of ML-based algorithms will have great benefit for improvement of personalized medicine in clinical practice and reduce mortality of CRC.

## Figures and Tables

**Figure 1 cancers-14-02045-f001:**
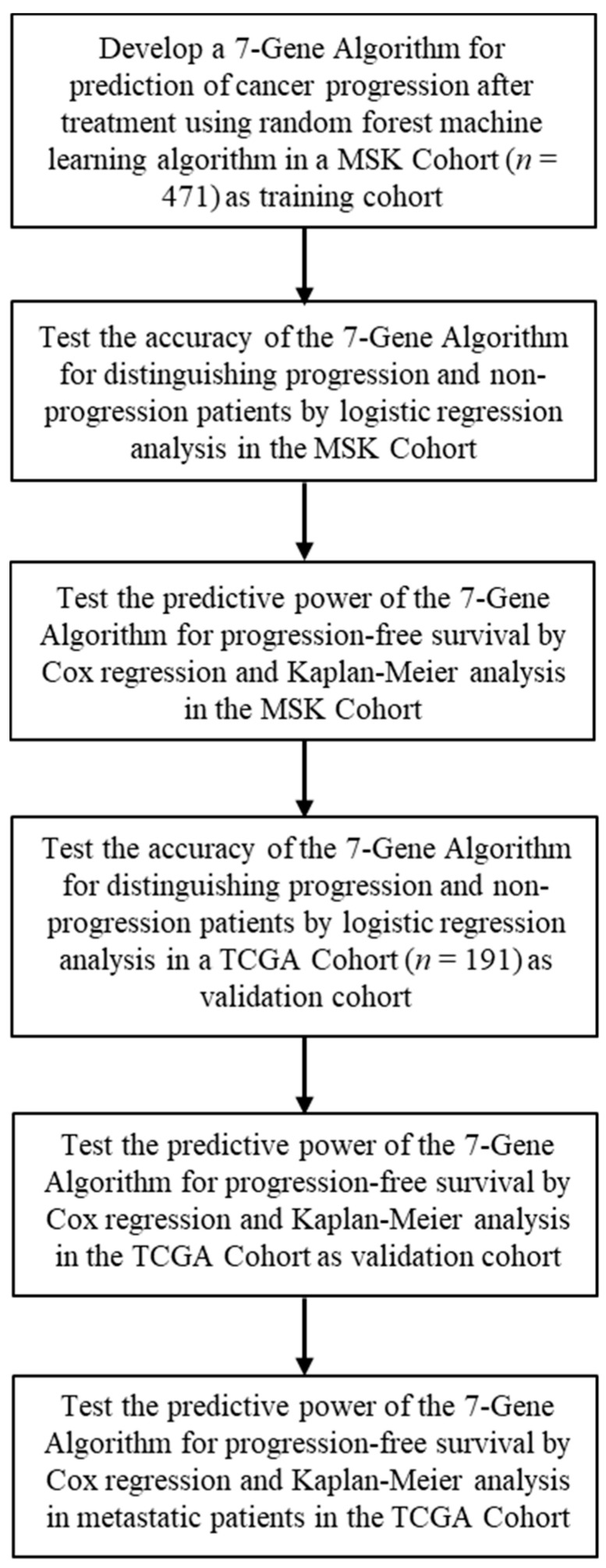
Study design.

**Figure 2 cancers-14-02045-f002:**
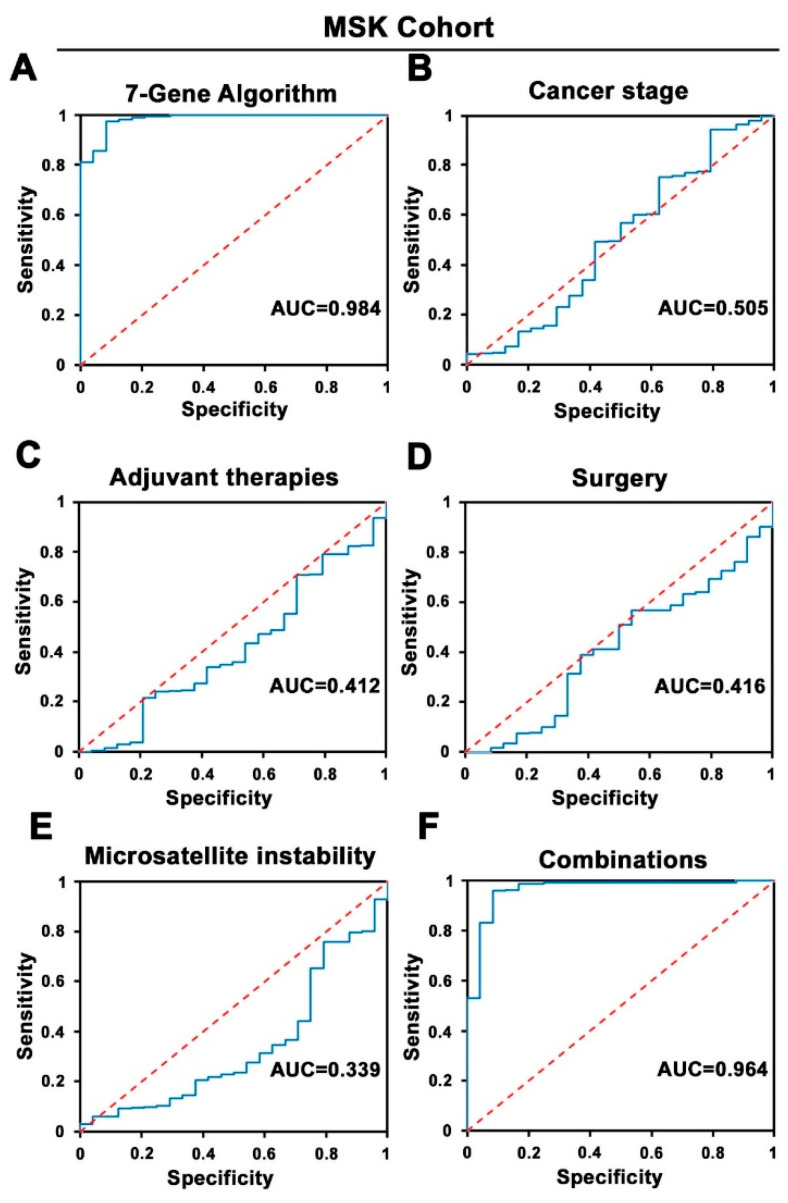
Receiver operating characteristic (ROC) curves of the 7-Gene Algorithm and clinical and pathological indicators for assessment of the performance accuracy for stratification of responder and non-responder group in the MSK cohort. (**A**) ROC curves of the 7-Gene Progression Algorithm. (**B**) Cancer stage. (**C**) Adjuvant therapies. (**D**) Surgery on primary tumor. (**E**) Microsatellite instability (MSI). (**F**) The 7-Gene Algorithm in combination with the parameters listed in (**B**–**E**). Sensitivity and specificity are indicated. The AUC values are indicated.

**Figure 3 cancers-14-02045-f003:**
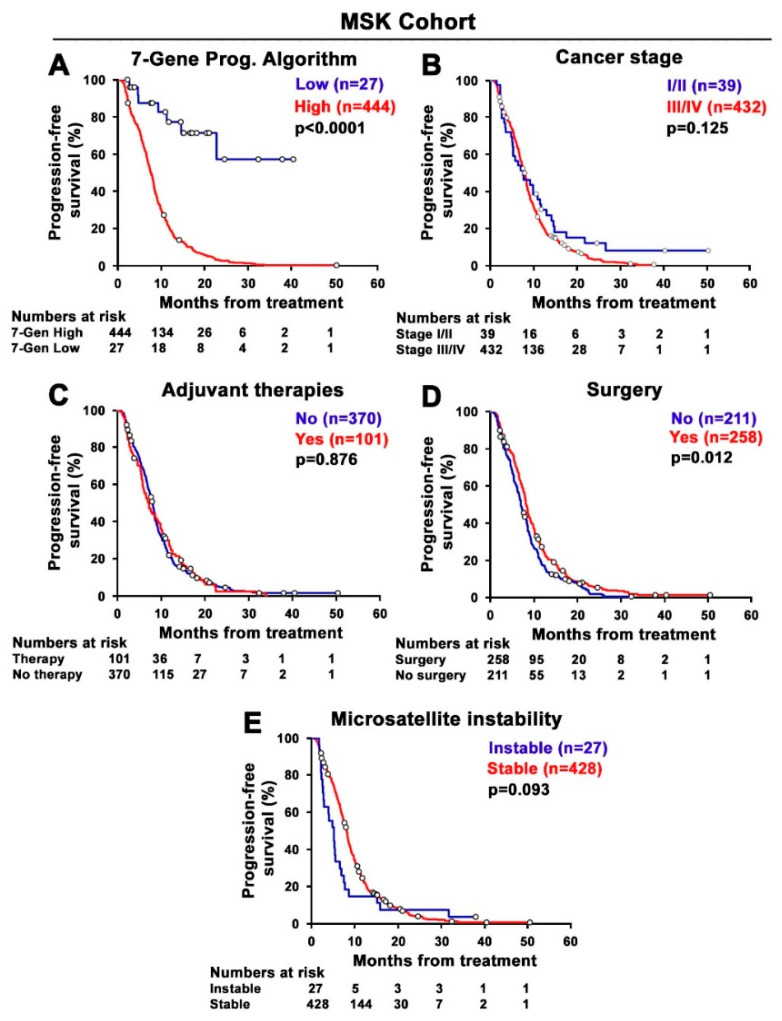
Kaplan–Meier survival analyses of the 7-Gene Algorithm and the clinical and pathological indicators for prediction of PFS in the MSK cohort. (**A**) The difference in PFS between two groups of CRC patients stratified based on the scores of the 7-Gene Algorithm. The statistical significance between the high and low group is indicated. (**B**) The difference in PFS between two groups of CRC patients stratified based on cancer stage. (**C**) Adjuvant therapies. (**D**) Surgery on primary tumor. (**E**) Microsatellite instability. Numbers of patients at risk in each time point are indicated.

**Figure 4 cancers-14-02045-f004:**
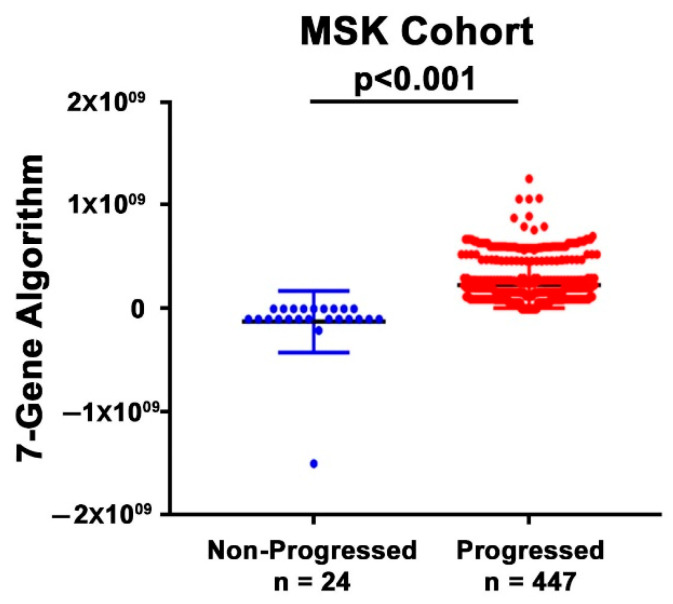
Dot plat analysis of the performance of the 7-Gene Algorithm as a classifier to distinguish subgroups of patients. Distribution of the scores of the 7-Gene Algorithm for responder (non-progression) and non-responder (disease progression) patients in the MSK cohort.

**Figure 5 cancers-14-02045-f005:**
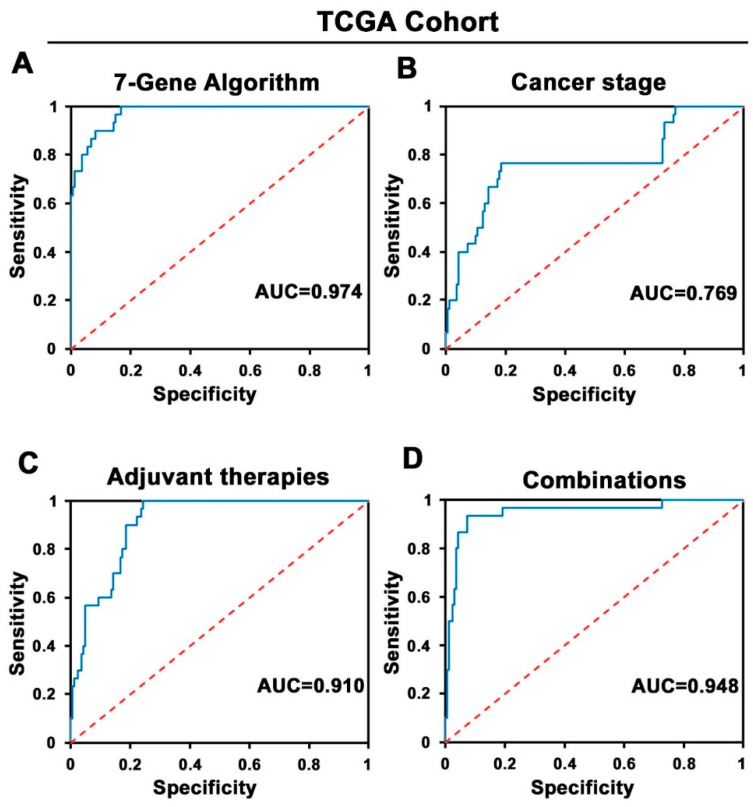
Receiver operating characteristic (ROC) curves of the 7-Gene Algorithm and the clinical indicators for assessment of the performance accuracy in stratification of responder and non-responder group in the TCGA Progression Cohort. (**A**) ROC curves of the 7-Gene Algorithm. (**B**) Cancer stage. (**C**) Adjuvant therapies. (**D**) The 7-Gene Algorithm in combination with all the above-mentioned clinical indicators. AUC values are shown.

**Figure 6 cancers-14-02045-f006:**
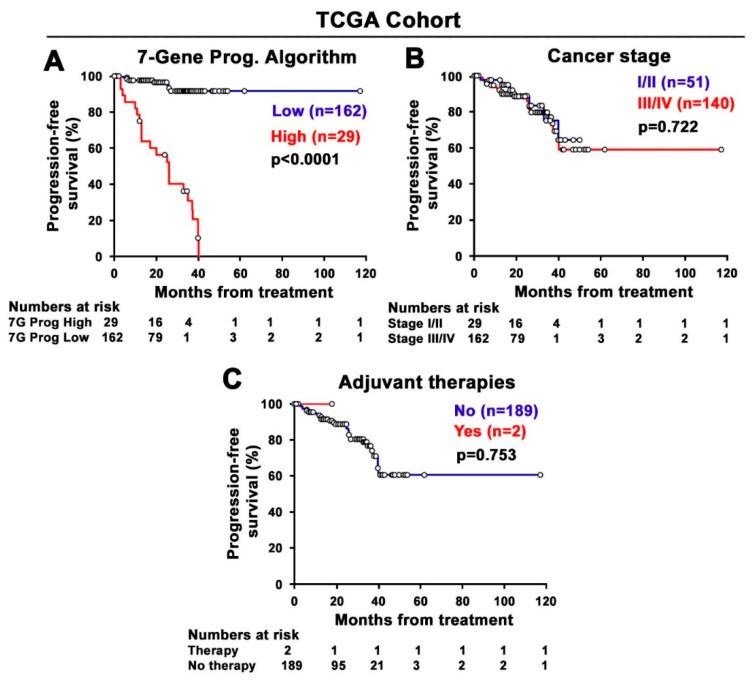
Kaplan–Meier survival analyses of the 7-Gene Algorithm and the clinical and pathological indicators for prediction of PFS in the TCGA cohort. (**A**) The difference in PFS between two groups of CRC patients stratified based on the scores of the 7-Gene Algorithm. The statistical significance between the high and low group is indicated. (**B**) The difference in PFS between two groups of CRC patients stratified based on cancer stage. (**C**) Adjuvant therapies. Numbers of patients at risk in each time point are indicated.

**Figure 7 cancers-14-02045-f007:**
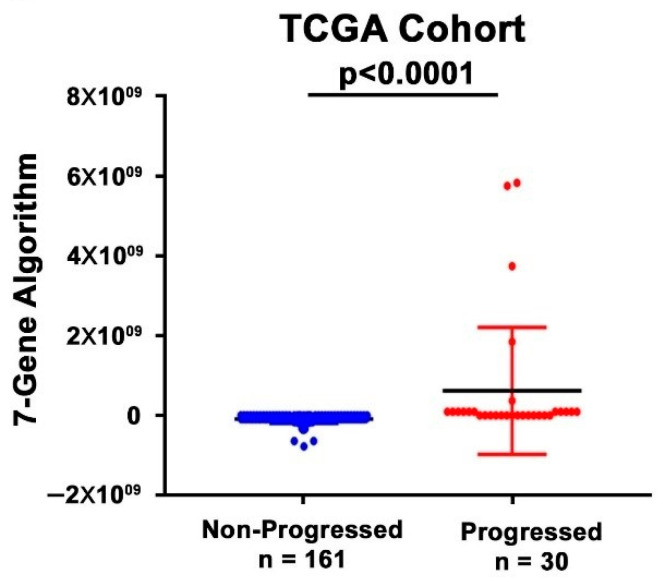
Dot plot analysis of the performance of the 7-Gene Algorithm as a classifier to distinguish subgroups of patients. Distribution of the scores of the 7-Gene Algorithm for responder (non-progression) and non-responder (disease progression) patients in the TCGA cohort.

**Figure 8 cancers-14-02045-f008:**
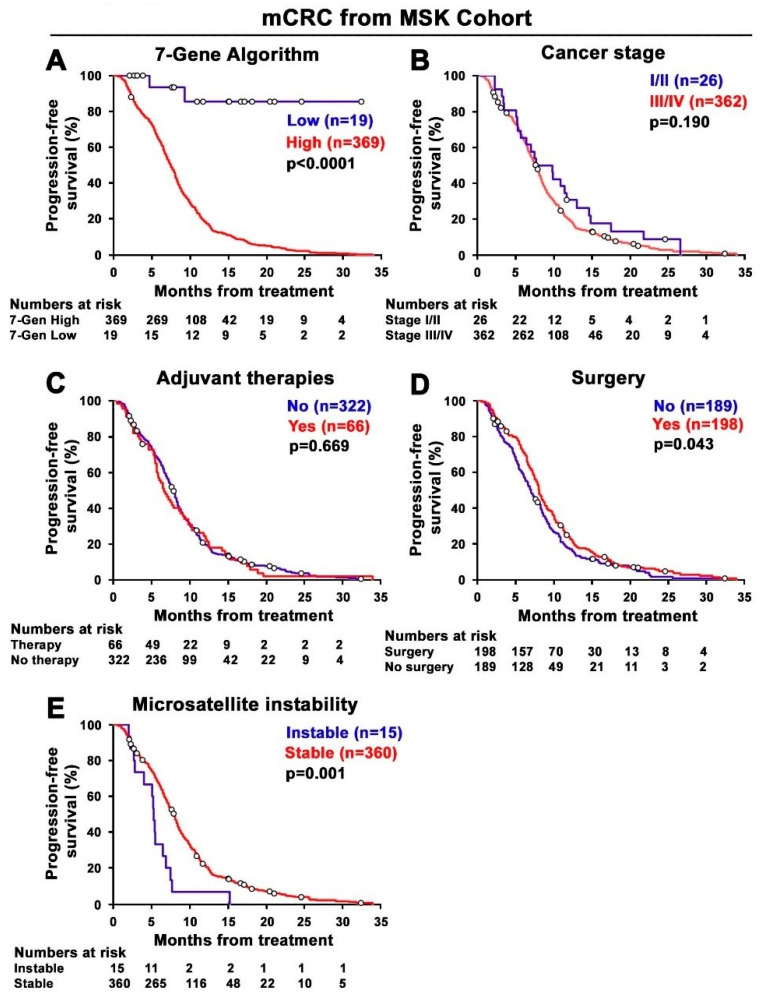
Kaplan–Meier survival analyses of the 7-Gene Algorithm and the clinical and pathological indicators for prediction of PFS in mCRC patients from the MSK cohort. (**A**) The difference in PFS between two groups of mCRC patients stratified based on the scores of the 7-Gene Algorithm. The statistical significance between the high and low group is indicated. (**B**) The difference in PFS between two groups of mCRC patients stratified based on cancer stage. (**C**) Adjuvant therapies. (**D**) Surgery on primary tumor. (**E**) Microsatellite instability. Numbers of patients at risk in each time point are indicated.

**Table 1 cancers-14-02045-t001:** Characteristics of the patients.

	MSK Cohort	TCGA Cohort
No of patients	471	191
Gender (Female) (%)	232 (49%)	92 (48%)
Gender (male) (%)	239 (51%)	99 (52%)
Median age (Q1, Q3)	59 (50, 68)	69 (62, 78)
Distant metastasis (%)	388 (82%)	21 (11%)
Cancers stage at diagnosis (%)		
Stage I	8 (2%)	8 (4%)
Stage II	31 (7%)	45 (24%)
Stage III	90 (19%)	125 (65%)
Stage IV	342 (73%)	13 (7%)
MSI type (%)		
Stable	428 (94%)	NA
Instable Prior adjuvant therapies (%)	27 (6%)	NA
Yes	370 (79%)	2 (1%)
No	101 (21%)	189 (99%)
Surgery on primary tumor (%)		
Yes	258 (55%)	NA
No	211 (45%)	NA
Overall survival (%)		
Living	160 (34%)	182 (95%)
Diseased	311 (66%)	9 (5%)
Progression/disease-free survival (%)		
Progressed	447 (95%)	161 (84%)
Non-progressed	24 (5%)	30 (16%)

MSI: microsatellite instability.

**Table 2 cancers-14-02045-t002:** Performance of the 7-Gene Algorithm and clinicopathological factors for distinguishing progression and non-progression after treatment in the MSK cohort (n = 471) and the TCGA progression cohort (n = 191).

	Sensitivity (95% CI)	Specificity (95% CI)	PPV (95% CI)	NPV (95% CI)
Prediction of Progression in the MSK Cohort (n = 471)				
7-Gene Algorithm	83% (68–98%)	98% (97–100%)	74% (58–91%)	99% (98–100%)
Cancer stage	0% (0–0%)	100% (100–100%)	0% (0–0%)	95% (93–97%)
Adjuvant therapies	0% (0–0%)	100% (100–100%)	0% (0–0%)	95% (93–97%)
Surgery on primary tumor	0% (0–0%)	100% (100–100%)	0% (0–0%)	95% (93–97%)
MSI	0% (0–0%)	100% (100–100%)	0% (0–0%)	95% (93–97%)
Combination	83% (68–98%)	99% (97–100%)	77% (61–93%)	99% (98–100%)
Prediction of Progression in the TCGA Progression Cohort (n = 191)				
7-Gene Algorithm	96% (93–99%)	77% (62–92%)	96% (93–99%)	79% (65–94%)
Cancer stage	100% (100–100%)	0% (0–0%)	85% (79–89%)	0% (0–0%)
Adjuvant therapies	100% (100–100%)	0% (0–0%)	84% (79–89%)	0% (0–0%)
Combination	96% (93–99%)	77% (62–92%)	96% (93–99%)	79% (65–94%)

CI: confidence interval; PPV: positive predictive value; NPV: negative predictive value; MSI: microsatellite instability.

**Table 3 cancers-14-02045-t003:** Univariate and multivariate Cox regression analyses of the 7-Gene Algorithm and clinicopathological factors for prediction of progression-free survival (PFS) in the MSK cohort (n = 471) and the TCGA cohort (n = 191).

	Univariate		Multivariate	
	HR (95% CI)	*p* Value	HR (95% CI)	*p* Value
**Prediction of PFS in the MSK Cohort (n = 471)**				
7-Gene Algorithm	7.5 (3.5–15.9)	<0.0001	8.9 (4.0–20.1)	<0.0001
Cancer stage	1.3 (0.9–1.9)	0.128	1.1 (0.7–1.5)	0.755
Adjuvant therapies	1.1 (0.8–1.3)	0.877	1.1 (0.8–1.4)	0.536
Surgery on primary tumor	0.8 (0–1.0)	0.013	0.7 (0–0.9)	0.002
MSI	0.7 (0.5–1.1)	0.097	0.6 (0–0.9)	0.009
**Prediction of PFS in the TCGA Cohort (n = 191)**				
7-Gene Algorithm	16.9 (7.2–39.6)	<0.0001	16.9 (7.2–39.7)	<0.0001
Cancer stage	1.2 (0.5–2.7)	0.723	1.3 (0.6–3.1)	0.539
Adjuvant therapies	3.0 × 10^−7^ (0-Inf)	0.997	1.7 × 10^−6^ (0-Inf)	0.996

HR: hazard ratio; CI: confidence interval; MSI: microsatellite instability.

**Table 4 cancers-14-02045-t004:** Univariate and multivariate Cox regression analyses of the 7-Gene Algorithm and clinicopathological factors for prediction of progression-free survival in mCRC patients (n = 388).

	Univariate		Multivariate	
	HR (95% CI)	*p* Value	HR (95% CI)	*p* Value
7-Gene Algorithm	16.9 (4.2–68.0)	<0.0001	17.6 (4.4–70.8)	<0.0001
Cancer stage	1.3 (0.9–2.0)	0.194	1.1 (0.7–1.7)	0.735
Adjuvant therapies	1.1 (0.8–1.4)	0.671	0.7 (0–1.6)	0.317
Surgery on primary tumor	0.8 (0–1.0)	0.044	0.7 (0–0.9)	0.003
MSI	0.4 (0–0.7)	0.002	0.4 (0–0.8)	0.003

HR: hazard ratio; CI: confidence interval; MSI: microsatellite instability.

## Data Availability

The data supporting reported results can be found in the publicly archived datasets analyzed or generated during the study.

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
