# Peer review of "Gene-Mutation-Based Algorithm for Prediction of Treatment Response in Colorectal Cancer Patients"

_cancers, 2022, doi:10.3390/cancers14082045_

Round 1
Reviewer 1 Report
I have read with great interest the paper by Heather Jonsson et al. “Gene mutations-based algorithms for prediction of treatment response in colorectal cancer patients”. In this manuscript, the authors describe a computational approach able to identify a mutational pattern in a group of 7 genes (KRAS, BRAF, ERBB2, MAP2K1, TSC2, TP53, and APC) useful as a tool to implement personalized therapy in advanced colorectal cancer. The paper is scientifically accurate and complete. Results are provide useful information to the field. The reference section is adequate, updated and appropriate to back up the points made in the article. The study conveys a clear-cut message. I have no major comments on this paper. However, it is necessary for the authors to make minor changes and corrections also to make the paper more attractive to readers:
The authors indicate on page 18 lane 104 that "Currently, no ML algorithms based on the mutation profiles of the significantly mutated genes that have been developed to stratify and predict disease metastasis and treatment response in CRC”. However, authors should also mention previous recent studies relevant to the topic such as “Choi JY et al. Clinical Implication of Concordant or Discordant Genomic Profiling between Primary and Matched Metastatic Tissues in Patients with Colorectal Cancer. Cancer Res Treat. 2020 Jul;52(3):764-778; Vera-Yunca D et al. Machine Learning Analysis of Individual Tumor Lesions in Four Metastatic Colorectal Cancer Clinical Studies: Linking Tumor Heterogeneity to Overall Survival. AAPS J. 2020 Mar 16;22(3):58; Chen PC et al. A Prediction Model for Tumor Recurrence in Stage II-III Colorectal Cancer Patients: From a Machine Learning Model to Genomic Profiling. Biomedicines. 2022 Feb 1;10(2):340.
Materials and Methods. When citing data extrapolated from datasets on the WEB, as in this case cBioportal, it is good practice to indicate the date of access and mining of the data.
I suggest expanding the discussion by taking as a starting point of comparison the results of two recent large mutational studies on some large cases of mCRCs (Antoniotti C et al. Tumour mutational burden, microsatellite instability, and actionable alterations in metastatic colorectal cancer: Next-generation sequencing results of TRIBE2 study. Eur J Cancer. 2021 Sep;155:73-84; Stahler A et al. Single-nucleotide variants, tumour mutational burden and microsatellite instability in patients with metastatic colorectal cancer: Next-generation sequencing results of the FIRE-3 trial. Eur J Cancer. 2020 Sep;137:250-259)
Author Response
Dear Editors and Reviewers,
We sincerely thank the Editors and Reviewers for your interests and valuable comments and suggestions concerning our manuscript. We are especially encouraged by the positive remarks provided by both Reviewers. In the revised manuscript, we have improved the Introduction section, Materials and Methods section, and the Discussion section. Your important suggestions have greatly helped us to improve the quality of our manuscript and finally meet the high standard requirements set by the Cancers. The revised text is marked in red. Below are our specific and point-by-point responses to address the important suggestions by Reviewers.
Response to Reviewer 1:
- We sincerely thank Reviewer 1 for providing information on these recent references! We have described these studies to highlight the importance of the relevant topics in the Introduction section, as suggested by Reviewer 1 (Lanes 105-114). We have included these references in the revised Reference list (Choi JY et al., 2020, Vera-Yunca D et al., 2020 and Chen PC et al., 2022).
- We sincerely thank Reviewer 1 for this important point! We have provided the date of access and mining of the data in the Materials and Methods section (Lane 154).
- We sincerely thank Reviewer 1 for this important suggestion! We have expanded the discussion by using the comparison of the two recent large mutational studies on some large cases of mCRC with our results to highlight the advantages of using comprehensive genomic profiles to develop prediction models. As suggested by Reviewer 1, we also used these recent studies which showed similar findings to support our findings in the Discussion section (Lanes 503-513). We have cited and included these two studies in the revised Reference list.

Reviewer 2 Report
The paper presents a machine learning approach to predict the treatment response in colorectal cancer patients.
The method is well written and explained in details and the results are interesting. I suggest to add some references to other approach of machine learning applied to cancers, such as SVM (P. Ferroni, F. M. Zanzotto, S. Riondino, N. Scarpato, F. Guadagni, and M. Roselli, “Breast cancer prognosis using a machine learning approach,” Cancers (Basel)., vol. 11, no. 3, pp. 1–9, 2019, doi: 10.3390/cancers11030328.) and decision tree (Jin, Seok Won, and Christina Soyoung Song. "Predicting adoption of colorectal cancer screening among Korean Americans using a decision tree model." Ethnicity & Health (2022): 1-15.)
Author Response
Dear Editors and Reviewers,
We sincerely thank the Editors and Reviewers for your interests and valuable comments and suggestions concerning our manuscript. We are especially encouraged by the positive remarks provided by both Reviewers. In the revised manuscript, we have improved the Introduction section, Materials and Methods section, and the Discussion section. Your important suggestions have greatly helped us to improve the quality of our manuscript and finally meet the high standard requirements set by the Cancers. The revised text is marked in red. Below are our specific and point-by-point responses to address the important suggestions by Reviewers.
Response to Reviewer 2:
We sincerely thank Reviewer 2 for providing excellent suggestions on including relevant studies to support our study aims. We have added and described these recent references in the Introduction section (Lanes 114-122). We have included these references in the revised Reference list (Ferroni P et al., 2019; Seok Won et al., 2022).
